# Molecular structure characterization analysis and molecular model construction of anthracite

Jinzhang Jia[1,2], Yumo Wu [1,2]*, Dan Zhao[3], Bin Li[1,2], Dongming Wang[1,2], Fengxiao Wang[1,2], Yinuo Chen[1,2]

**1** College of Safety Science and Engineering, Liaoning Technical University, Fuxin, Liaoning, China, **2** Ministry of Education, Key Laboratory of Mine Thermal Power Disaster and Prevention, Fuxin, Liaoning, China, **3** Faculty of Civil Engineering and Architecture, Zhanjiang University of Science and Technology, Zhanjiang, Guangdong, China

\* 13614067811@163.com

## Abstract

Coal is the largest non-renewable energy as well as an important basic energy and industrial raw material. Thus, correctly understanding the molecular structure characteristics of coal has important theoretical value for realizing carbon neutralization. In this work, we clarified the molecular structure characteristics of anthracite, where the organic matter in anthracite was characterized and analyzed by industrial/elemental analysis, FTIR, XPS, XRD and solid $^{13}$C NMR. The ratio of bridge carbon to the perimeter carbon of anthracite was 0.38, and the degree of condensation in the aromatic structure was high. Nitrogen in coal primarily exists in the form of pyridine and pyrrole. Based on the information on functional group composition, the carbon skeleton structure, and surface element composition, a molecular structure model of Yangquan anthracite could be constructed, where the molecular formula was $C_{208}H_{162}O_{12}N_4$. This study may serve as a reference for researchers in this field to consult and refer to the construction ideas and methods of molecular structure models of different coal samples.

## 1. Introduction

Coal is an indispensable fuel energy resource in many countries. and exploring the detailed chemical information such as the structural characteristics of coal organic matter can efficiently transform and utilize coal resources. Unlike other organic polymers, coal has no uniform physical and chemical forms, with various molecular compositions and complex chemical structures, making it difficult to systematize the molecular structure of coal. As a high-quality coal, anthracite shows these chemical characteristics in particular [1]. Due to the high coalification degree of anthracite, its combustion and pyrolysis processes have been widely used in industrial production. Therefore, the establishment of an efficient and accurate comprehensive method to construct the molecular structure of coal offers considerable value for studying the characteristics and mechanism of coal from the microscopic level.

The coal molecular structure model consists of model developed gradually in the field of coal science over the past 70 years. After continuous improvement and development, the coal

---

**Data Availability Statement:** All relevant data are with in the paper and its Supporting Information files.

**Funding:** This work was partly supported by the National Natural Science Foundation of China

---

(grant number 52174183), and the Natural Science Foundation of Liaoning Province (grant number 2019-MS-162). All funders are the first author Jia Jinzhang.

**Competing interests:** The authors have declared that no competing interests exist.

molecular structure model has gone through the Fuchs model, Given model, Wiser model, and Shinn model in turn [2–5]. With the development of precision instruments, the recognition range and precision of modern analytical instruments have greatly improved. Zhao et al. [6] used in situ Fourier transform infrared (FTIR) to analyze the surface characteristics of coal and obtained four types of active functional groups, where Okolo et al. [7] was tested by Fourier transform infrared spectroscopy (ATR-FTIR) and solid-state $^{13}C$ nuclear magnetic resonance spectroscopy, and the chemical structural characteristics of different rank coals were obtained. Meng et al. [8] established a molecular structure model of coal based on FTIR and X-ray photoelectron spectroscopy (XPS)techniques and explored the low-temperature oxidation reaction of coal. The results showed that the low-temperature oxidation reactant of coal had α carbon atoms, hydroxyl, and ether groups on the aromatic ring, besides the active aliphatic chain. Marcano et al. [9] determined the size distribution of aromatic rings in coal based on the lattice fringes of multiple high-resolution transmission electron microscopes and generated a large-scale molecular model of coal in the molecular modeling space. Leyssale et al. [10] used two-dimensional high-resolution transmission electron microscopy (HRTEM) lattice fringe image analysis, three-dimensional image synthesis, and atomic simulation to construct a pyrolysis carbon atomic model based on the nano-structure characteristics of the large-scale total carbon atomic model obtained from the HRTEM image of pyrolysis carbon. The distribution function was verified according to the experiment, showing good consistency. Saikia et al. [11] obtained the number of layers of the two coal samples and the average number of carbon atoms per aromatic graphene by measuring the random layered structural parameters of coal using the X-ray diffraction technique. Niekerk et al. [12] assembled a single molecule into a three-dimensional structure by adding sulfur, nitrogen, oxygen and aliphatic side chains and crosslinked bonds to the aromatic skeleton based on $^{13}C$ nuclear magnetic resonance spectroscopy (NMR); thus, establishing a molecular model of two South African coals. Baysal et al. [13] obtained the characterization information of aliphatic, aromatic, and functional groups in coal by FTIR, $^{13}C$ NMR, and X-ray diffraction (XRD) to construct the macromolecular structure model of coal, and the aromaticity values obtained by XRD and $^{13}C$ NMR showed high correlation. Yu et al. [14] constructed a large-scale molecular model of anthracite using $^{13}C$ NMR, XRD, and XPS techniques to capture the pore orientation caused by coalification in anthracite, while Mokone et al. [15] utilized petrographic analysis, elemental analysis, helium density, $^{13}C$ NMR, XRD and HRTEM. The experimental data were used to construct the molecular structures of polycyclic aromatic hydrocarbons containing oxygen, nitrogen, and sulfur elements, which provided a better and more accurate measurement of aromaticity and bone density. Cui et al. [16] used $^{13}C$ NMR nuclear magnetic resonance spectroscopy, attenuated total reflection FTIR, and quantitative chemical analysis to analyze the carbon skeleton structure and coal structural characteristic parameters of anthracite. In addition, the aromaticity, hydrogen aromaticity, and average aromatic nucleus size of anthracite, were calculated to construct a molecular structure model of anthracite. According to the above analysis, the molecular modeling method showed great potential in the study and could be better used to understand the molecular structure of coal. According to current worldwide research, the construction methods of coal molecular models have not been unified, and the accuracy and applicability of model construction must be further verified, especially whether a constructed coal molecular model can truly reflect the molecular structure of an experimental coal sample.

Thus, in this work, we determined the content of C, H, O, N and S in coal by elemental analysis, where the aromatic structure, fat structure and oxygen-containing functional groups in the coal were revealed by $^{13}C$ NMR and FT-IR. In addition, XPS was used to reveal the occurrence state of N and S elements in coal, and the microcrystalline structure of coal was

**Table 1. Proximate analysis and ultimate analysis of Yangquan anthracite.**

| Proximate analysis(wt%) | | | Ultimate analysis(wt%,daf) | | | | |
|---|---|---|---|---|---|---|---|
| Moisture on an air-dried basis($M_{ad}$) | Ash on a dry basis($A_{ad}$) | Volatile matter on a dry and ash-free basis($V_{ad}$) | C | H | O | N | S |
| 1.70 | 13.35 | 7.44 | 85.53 | 4.52 | 6.53 | 1.98 | 0.44 |

revealed by XRD. Subsequently, according to the above characterization results, the molecular structure model of Yangquan anthracite was constructed. On this basis, we used Materials Studio software to compare the adsorption data of the $CH_4$ molecules on the molecular model of anthracite with the experimental data of anthracite from reference [17], which proved the rationality of the molecular model of anthracite established in this paper.

## 2. Experimental

### 2.1 Proximate and ultimate analysis of anthracite

Proximate analysis and ultimate analysis of coal provide the basic content of coal quality analysis, and also offer an important link for the construction of coal macromolecular structures [18]. Anthracite samples were obtained from the Yangquan Coal Mine, Shanxi Province, China. Anthracite samples with particle size of more than 200 mesh were prepared by repeatedly crushing, screening, and shrinking the coal samples with the air-dried anthracite using the raw coal, a crusher, and a vibrating screen. Then, the moisture content was measured and recorded when 100 g of coal sample was heated to 105˚C. $N_2$ was added at 845˚C and the ash content was determined. Then, the temperature was adjusted to 900˚C to keep the temperature constant, the volatile matter of the coal samples was determined, and the coal sample was decomposed by catalytic oxidation in the oxygen environment at 1150˚C. The elements in the coal sample were transformed into oxides, which entered into the adsorption column, and the gas was separated by the adsorption-desorption column and then entered into the thermal conductivity cell detector for ultimate analysis and detection. The results of proximate and ultimate analysis of anthracite are shown in Table 1.

### 2.2 Sample characterization

**2.2.1 FTIR analysis.**  The position and intensity of the infrared absorption peaks can be related to the molecular composition or functional group content [19]. The spectral range recorded by the FTIR spectrometer was 4000–400cm$^{-1}$, where the moving mirror speed was 0.4747, and the resolution was 0.04cm$^{-1}$. The test sample was mixed with 0.2 g of potassium bromide in 0001 g of pulverized coal with a particle size of 400 mesh, which was fabricated into transparent thin slices with a thickness of 0.2–0.5 mm. The infrared spectra of the anthracite coal samples were obtained after 32 scans. The original FT-IR spectrum of anthracite consisted of the transmittance-wavenumber spectrum. To facilitate the quantitative analysis of the content of functional groups in the coal samples, the transmittance and absorbance were transformed by the Lambert–Beer law [20], which could be expressed by:

$$A = \lg(1/T) \tag{1}$$

where $A$ is absorbance Abs. Unit; $T$ denotes transmittance (%).

**2.2.2 X-ray photoelectron spectroscopy.**  As a surface structure analysis technology, X-ray photoelectron spectroscopy can directly determine the occurrence state and relative content of elements. It offers high sensitivity and reliability and has been widely used in the study of coal structures [21, 22]. The XPS full-spectrum scanning parameters of anthracite were as

follows: X-ray source, voltage of 16 KV, current of 14.9 mA, a beam spot diameter of 650 um, pass energy of 100 eV, and elemental high-resolution spectrum of 30 eV, where the charge calibration was based on C1s = 284.4 eV. In this paper, nitrogen and sulfur in anthracite were analyzed to obtain more accurate content information of the surface functional groups of nitrogen and sulfur. At the same time, Origin software was used for peak fitting to analyze the different states of the nitrogen and sulfur atoms.

**2.2.3 XRD analysis.** The XRD test conditions consisted of: Cu target, K radiation, tube current of 30 mA, a divergent slit of 1 mm, receiving slit of 0.30 mm, step scanning, step width of 0.029, scanning speed of 2%/min, and scanning range 10° to 90°(2θ). The diffraction patterns were fitted by Origin software, and the peak position, intensity, FWHM, and peak area were determined.

A certain number of small crystals will be found in the molecular structure of coal, which belong to a type of organic matter with short-range order but with the long-range disorder. These tiny crystals in the molecular structure will be stacked in different parallel degrees by several aromatic rings. Combined with the parameters in the diffraction spectrum, the microcrystalline structure parameters of coal can be obtained according to the Bragges and Scherrer Formula (2) [23–25].

$$
\begin{cases}
d_{002} = \dfrac{\lambda}{2\sin\theta_{002}} \\[2ex]
L_a = \dfrac{1.84\lambda}{\beta_{100}\cos\theta_{100}} \quad L_c = \dfrac{0.94\lambda}{\beta_{002}\cos\theta_{002}} \\[2ex]
f_a = \dfrac{A_{002}}{A_{\gamma} + A_{002}} \qquad N_{ave} = \dfrac{L_c}{d_{002}}
\end{cases}
\tag{2}
$$

where $\lambda$ denotes the X-ray diffraction wavelength was 0.154 nm; $\theta_{002}$ is the peak weighted average center of 002 peaks; $\theta_{100}$ is the peak weighted average center of 100 peaks; $\beta_{002}$ is the half-peak width of 100 peaks; $\beta_{100}$ is the half-peak width of 100 peaks; $A_{\gamma}$ is the $\gamma$ peak area; $A_{002}$ is the 002 peak area.

**2.2.4 Solid-state [13]C NMR.** [13]C NMR can be used to quantitatively and qualitatively analyze the structure and composition of the organic materials, where the spectra consist of usually simple single peaks, and the position of the peak will determine the corresponding chemical shift, which can be used to directly obtain the information of the carbon skeleton of coal [26]. To obtain the ideal spectrum, cross polarization, magic angle rotation, and TOSS sideband suppression techniques were used, where the contact time was 3 ms, the spectral width was 30000 Hz, the MAS spin rate was 10 kHz, the nuclear magnetic resonance frequency was 75.47 MHz, the cycle delay was 5 s, and the scanning number was 2000–4000.

The main coal macromolecular skeleton was composed of carbon atoms in the molecular structure of coal, where the other groups were connected to the carbon skeleton in different ways. The chemical shifts of the [13]C NMR peaks corresponding to different types of carbon atoms (aliphatic carbon, aromatic carbon) or different functional groups connected to them were also different [27]. According to the chemical shifts, the peak positions and content percentage of the different carbon atoms, with 12 specific structural parameters of the anthracite coal dust structure, could be obtained [28], as shown in Table 2.

The ratio of bridge carbon to peripheral carbon ($X_{BP}$) in anthracite could be obtained by calculations, which could be used to characterize the condensation degree of the aromatic structure in the coal macromolecular structure [29, 30]. The following calculation formula was

**Table 2. Denotations of the coal structure parameters.**

| Parameters | Denotations |
|---|---|
| $f_{al}^{*}$ | Content of aliphatic methyl and aromatic methyl carbon |
| $f_{al}^{H}$ | Content of quaternary carbon, CH and $CH_2$ group carbon |
| $f_{al}^{O}$ | Content of oxygen-bonded carbon |
| $f_{a}^{H}$ | Content of protonated aromatic carbon |
| $f_{a}^{B}$ | Content of aromatic bridgehead |
| $f_{a}^{S}$ | Content of alkyl-substituted aromatic carbon |
| $f_{a}^{P}$ | Content of phenolic hydroxyl or ether oxygen-bonded carbon |
| $f_{a}^{N}$ | Content of non-protonated aromatic carbon |
| $f_{a}^{C}$ | Content of carbonyl |
| $f_{al}$ | Aromatic carbon ratio |
| $f_{a}$ | Total content of aromatic carbon |
| $f_{a}^{'}$ | Content of aromatic nucleus carbon |

used:

$$X_{BP} = \frac{f_a^B}{f_a^H + f_a^P + f_a^S} \quad (3)$$

# 3. Results and discussion

## 3.1 Fourier transform infrared spectroscopy analysis of anthracite

The spectrum transformed by Formula (1) is shown in Fig 1, where the 3600–3000 cm$^{-1}$ band was called the hydroxyl absorption band, the 3000–2800 cm$^{-1}$ band was called the aliphatic hydrocarbon absorption band, the 1800–1000 cm$^{-1}$ band was the oxygen-containing functional groups and partial aliphatic hydrocarbon absorption band, and the aromatic absorption band was 900–700 cm$^{-1}$.

**3.1.1 Absorption band analysis of the aromatic structure.** Aromatic rings are the main structures in coal and the main carrier of gas adsorption by coal [31]. A study on the aromatic structure of coal samples could be used to explore the differences in the gas adsorption capacity of coal samples with different metamorphic degrees, and also provide a theoretical basis for the construction of molecular models and adsorption simulation. The peak analysis module in the Origin software was used to fit each spectrum of anthracite in the 900–700 cm$^{-1}$ band, where the fitting model was Gaussian, as shown in Fig 2. The fitting results showed that the main absorption peaks of the aromatic structure in the three different coal samples were near 870 cm$^{-1}$, 800 cm$^{-1}$, and 750 cm$^{-1}$. Among them, the peak functional groups near 870 cm$^{-1}$ consisted of the benzoyl penta-substituted, the peak functional groups near 800 cm$^{-1}$ were trisubstituted or tetrasubstituted benzene rings, and the peak functional groups near 750 cm$^{-1}$ were the disubstituted benzene ring.

Based on peak-dividing fitting, the peak-dividing parameters of the infrared absorption peak of anthracite in the 900–700 cm$^{-1}$ band were calculated and analyzed. We found that the peak area of the benzene ring pentasubstituted in anthracite was 2.060, where the relative area ratio was 33.967%. The peak area of the benzene ring trisubstituted or tetrasubstituted was 2.177, the relative area ratio was 35.896%, and the peak area of the benzene ring disubstituted was 1.827, and the relative area ratio was 30.134%. This indicated that the macromolecular structure of anthracite was dominated by benzene ring tri-substituted, benzene ring tetra-substituted, or benzene ring penta-substituted, and supplemented by benzene ring di-substituted.

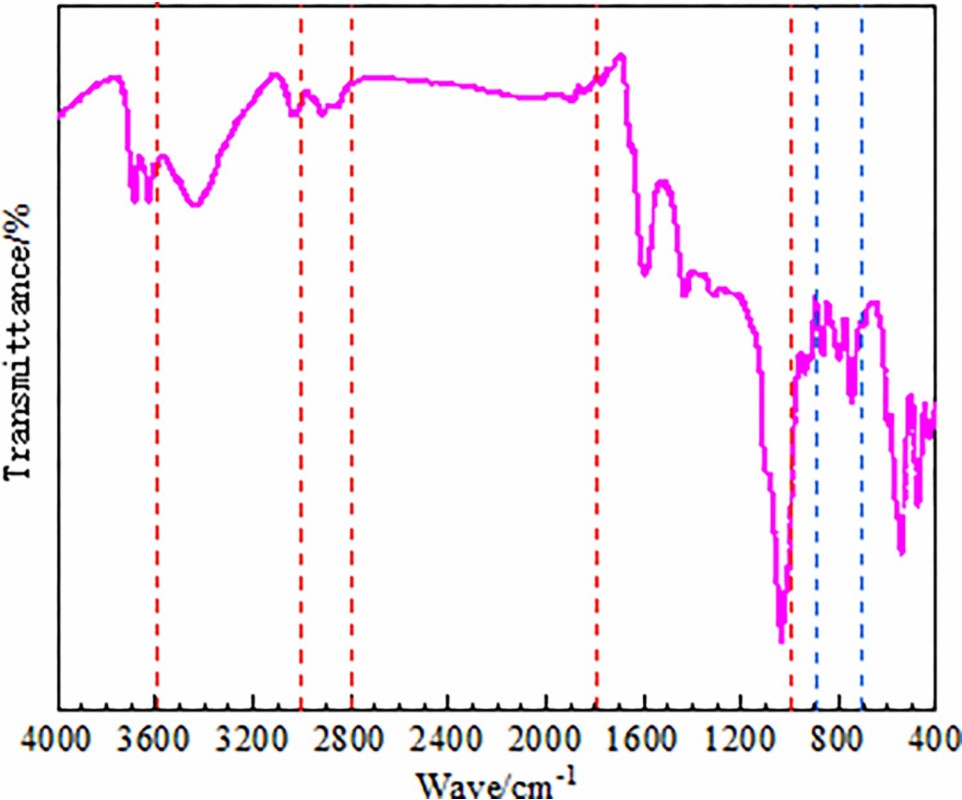

**Fig 1. FT-IR spectra of anthracite.**

**3.1.2 Analysis of absorption bands of oxygen-containing functional groups.** The oxygen-containing functional groups in coal mainly included hydroxyl (-OH), carboxyl (-COOH), carbonyl (C = O), and ether oxygen bonds (R-O-R′). In addition to the absorption peak of the oxygen-containing functional groups, we observed stretching vibrations in the aromatic carbon-carbon double bonds (C = C), and deformation vibrations of the methyl (-CH$_3$) and methylene (-CH$_2$) groups in the infrared wavenumber 1800–1000 cm$^{-1}$ region, where the spectrum in this region was more complex. The absorption peaks in the band of 1000–1800 cm$^{-1}$ in the infrared spectrum of anthracite were fitted by peak separation, as shown in Fig 3.

The fitting results showed that the carboxylic acid content of anthracite was only 0.51%, while in the oxygen-containing functional groups of anthracite in this region, the ratio of phenol, alcohol, ether (C-O) to carboxyl, and carbonyl (C = O) was 17.4:1. This showed that the content of oxygen-containing functional groups in the coal samples with a high degree of coalification was low.

**3.1.3 Analysis of aliphatic hydrocarbon absorption band.** In the FT-IR spectrum of the coal, the wavenumber range of 2800–3000 cm$^{-1}$ belonged to the absorption range of -CHx in the lipid chain and lipid ring. The spectrum of anthracite in this band was fitted, as shown in Fig 4. The infrared spectrum of this band contained three absorption peaks, two main absorption peaks (near 2850 cm$^{-1}$ and 2920 cm$^{-1}$) and a weak absorption peak (near 2950 cm$^{-1}$). The region near peak 2950 cm$^{-1}$ was attributed to the asymmetric -CH$_3$ stretching vibration, while the absorption peak near 2920 cm$^{-1}$ reflected the asymmetric -CH$_2$ stretching vibrations of the aliphatic hydrocarbons or naphthenic hydrocarbons in coal. The absorption peak near the 2850 cm$^{-1}$ peak represented the symmetric stretching vibrations of -CH$_2$.

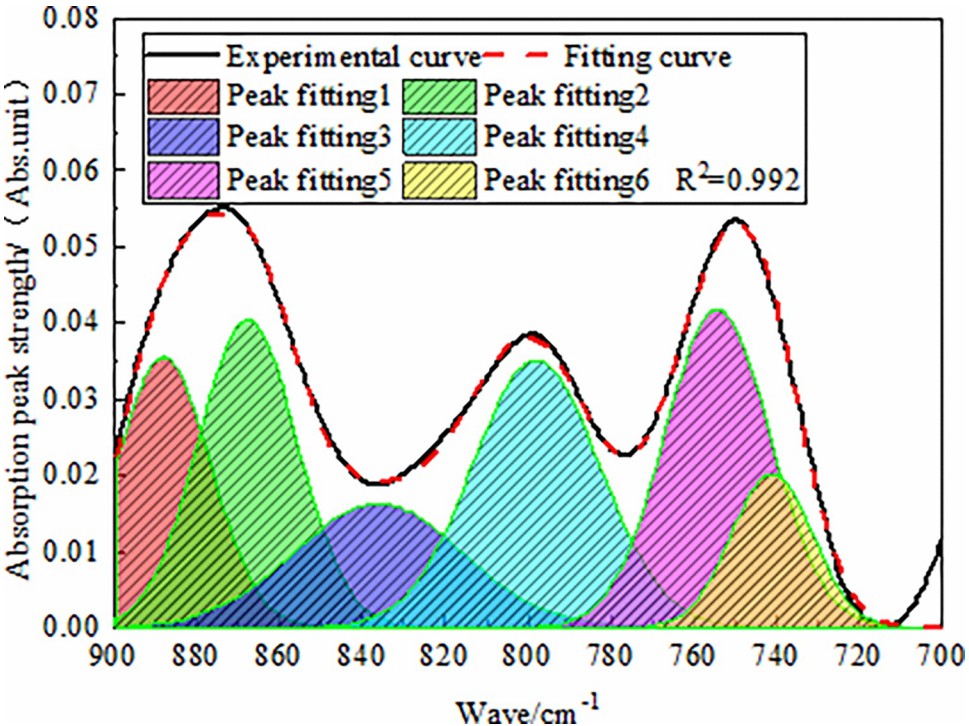

**Fig 2. FT-IR fitting spectra of anthracite in 900–700 cm$^{-1}$ band.**

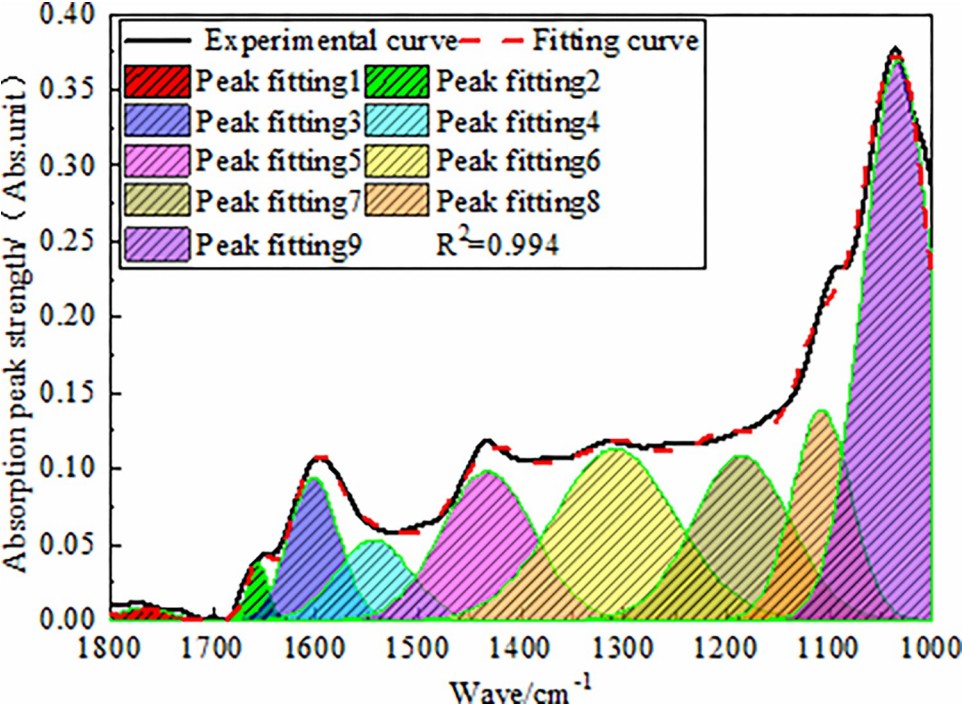

**Fig 3. FT-IR fitting spectra of anthracite in 1800–100 cm$^{-1}$ band.**

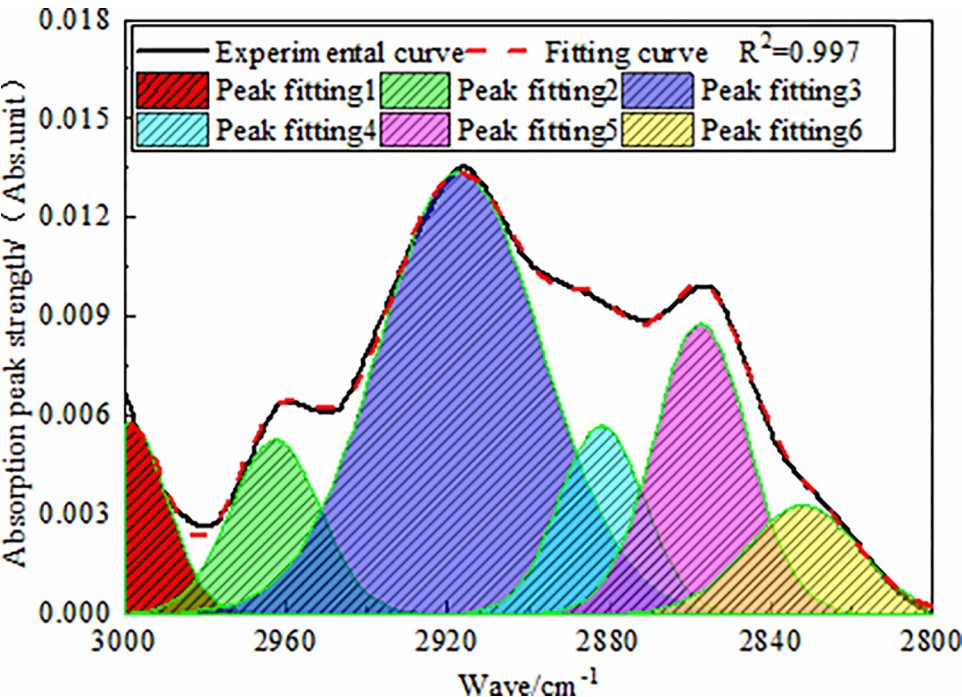

**Fig 4. FT-IR fitting spectra of anthracite in 3000–2800 cm$^{-1}$ band.**

As shown by the fitting results, the aliphatic hydrocarbon content of anthracite was mainly dominated by the stretching vibrations of asymmetric $CH_2$ (relative area: 47.64%), supplemented by the stretching vibrations of symmetric $CH_2$ (relative area: 26%), indicating that the aliphatic chains in the molecular structure of anthracite sample were mainly short-chain structures.

In addition, the fat hydrocarbon content of anthracite was 1.416, indicating that anthracite had a strong adsorption capacity for coalbed methane.

**3.1.4 Analysis of hydroxyl absorption band.** Hydroxyl groups are the main functional groups that form hydrogen bonds in coal, while hydrogen bonds act as very important secondary bonds in coal macromolecular structures, playing an extremely important role in the association and destruction of the macromolecular structure network. In the infrared spectrum of anthracite, the wavenumber was in the range of 3000–3600 cm$^{-1}$, belonging to the hydroxyl absorption band. The peak fitting results of anthracite in this region are shown in Fig 5. The assignment of different peak positions showed that the hydroxyl structure mainly included OH-N hydrogen, ring hydrogen, OH-ether hydrogen, OH-OH hydrogen, and OH-π hydrogen bonds.

As shown by the fitting results, the total absorption peak area of hydroxyl in anthracite was 17.777. The total hydroxyl groups in anthracite were mainly provided by the OH-OH hydrogen bonds (relative area 32.83%) and OH-π hydrogen bonds (32.60%), followed by OH-O hydrogen bonds (15.26%) and cyclic hydrogen bonds (13.77%), where the content of OH-N hydrogen bonds (5.54%) was relatively small.

## 3.2 Occurrence of nitrogen and sulfur in XPS fitting analysis

The XPS full spectrum scan of anthracite is shown in Fig 6. In this work, nitrogen and sulfur elements in the coal samples were analyzed to obtain the occurrence state and content information of the nitrogen and sulfur elements.

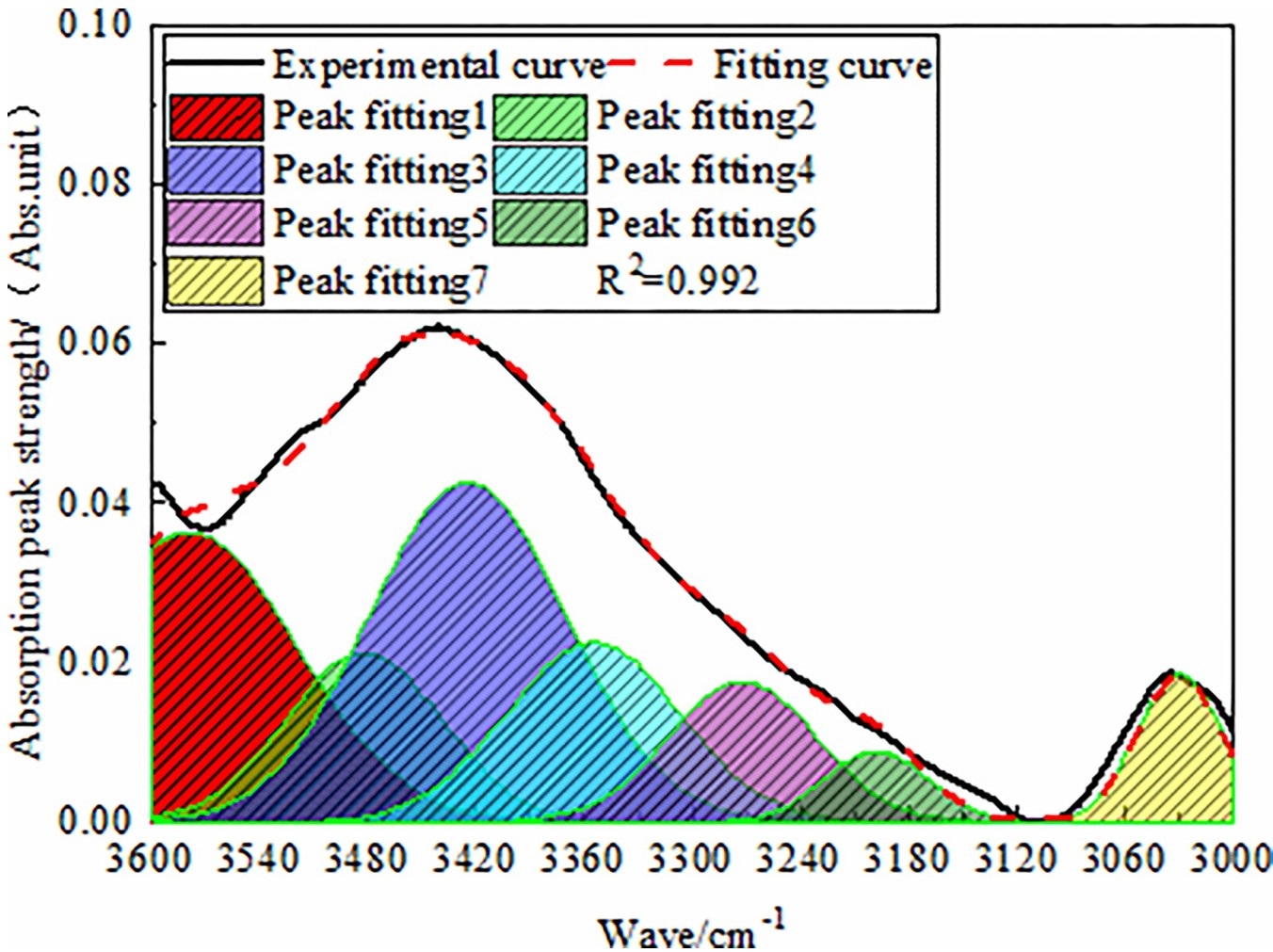

**Fig 5. FT-IR fitting spectra of anthracite in 3600–3000 cm$^{-1}$ band.**

The four characteristic peaks of nitrogen in coal were fitted by XPS analysis, which consisted of pyridine nitrogen ($C_5H_5N$), and the binding energy of the peak position was 398.8 ± 0.4 eV. In pyrrole nitrogen ($C_4H_5N$), the peak position binding energy was 400.2 ± 0.3 eV, while in seasonal nitrogen-N- $(CH_3)_3$, the peak position binding energy was 401.4 ± 0.3 eV, and in nitrogen oxide ($N_xO_y$), the peak position binding energy was 402.9 ± 0.5 eV. When XPS was used to analyze the sulfur forms in coal, most could be divided into four categories, namely thiols, sulfides, thiophenes, sulfones, sulfoxides, and inorganic sulfur. The distribution ranges of the electronic binding energy were 162.2–164, 164–164.4, 165–168, and 169–171 eV. The XPS data fluctuated greatly; thus, the first data were smoothed, and then peak fitted [32], for the anthracite nitrogen and sulfur peak fitting results, as shown in Fig 7.

According to the fitting results, pyridine nitrogen (25.24%) and pyrrole nitrogen (41.28%) in anthracite were the main occurrence forms of nitrogen in coal. The reason was that pyrrole nitrogen and pyridyl nitrogen were derived from chloroplasts and alkaloids of coal-forming plants, respectively. These were both aromatic conjugated systems with high stability; thus, they were stably preserved in the coal-forming process. Nitrogen oxide (17.36%) was mainly produced by the oxidation of pyridine nitrogen and pyrrole nitrogen in the air, with less relative content. The content of quaternary nitrogen (16.12%) was the lowest, as the −CH$_3$ in the

**Fig 6. XPS spectra of anthracite.**

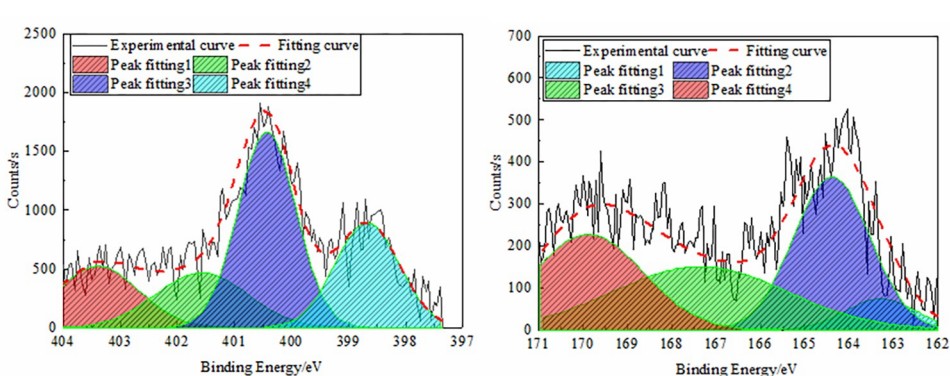

**Fig 7. Peak fitting spectra of nitrogen and sulfur in anthracite.** (a) Nitrogen, (b) Sulfur.

coal structure was relatively exfoliated during coal metamorphism, and the aliphatic hydrocarbons formed long-chain or ring structures. At the same time, the aromatic structure could undergo ring condensation of dehydrogenation. The main form of sulfur in the anthracite molecular structure was thiophene (35.34%), sulfone, sulfoxide (32.30%), and inorganic sulfur (25.97%) where the auxiliary, mercaptan, sulfide (6.39%) content was the lowest. The reason was that in the process of coalification, the thermal effect caused most of the functional groups such as sulfides and thiols to produce conversion or loss, causing the molecular consistency to increase and the structure to become more stable. Due to the particularity of the thiophene ring with the aromatic conjugated structure, it was one of the products of the unstable side chain sulfur speciation transformation, and thiophene gradually became the main organic sulfur structure.

### 3.3 Structural parameters from XRD analysis

In this work, the change characteristics of the microcrystalline structure of anthracite were analyzed by the changes in the XRD spectral structure parameters, and the XRD spectrum of anthracite is shown in Fig 8A. There were two broad peaks in the XRD spectra of the coal samples in the ranges of diffraction angles of $2\theta = 20°–30°$ and $2\theta = 40°–50°$, which belonged to the XRD peaks of organic matter. Among them, $2\theta = 20°–30°$ corresponded to the 002 surfaces of the microcrystalline structure, which was called the 002 peak, and was the characteristic peak reflecting the $L_c$ and $d_{002}$ values. In this case, the sharper the peak shape, the closer the peak position θ was to the left, indicating that the metamorphic degree of coal samples was deeper. At the same time, it also indicated that the proportion of aromatic slices representing the ordered structure in coal increased, indicating that the orientation of aromatic slices was better at the micro-level. Furthermore, the γ band reflected the side-chain structure (aliphatic branched chains, functional groups, and aliphatic hydrocarbons) connected to the ordered microcrystalline, which reflected the disordered structure in coal to some extent [33]. The 002 peak in the coal samples was related to the distance between the aromatic ring layers, and the band was related to the aliphatic groups (including the aliphatic side chains and aliphatic rings) in the molecular structure. At $2\theta = 40°–50°$, the corresponding peak consisted of 100 peaks of the microcrystalline structure, which characterized the condensation degree of

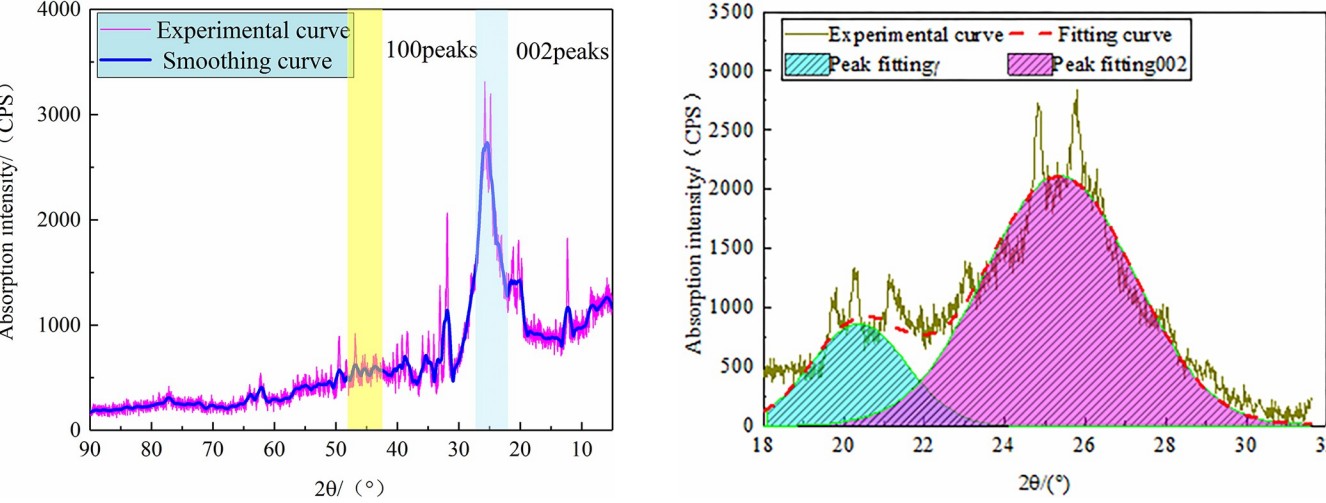

**Fig 8. XRD spectrogram.** (a) X-ray diffractogram of the anthracite coal sample, (b) XRD-002 Peak Fitting Map of Anthracite.

**Table 3. XRD structural parameters of anthracite coal.**

| Structural parameters | Interlayer distance ($d_{002}$/nm)) | Extension of aromatic laminates($L_a$/nm) | Stacking size ($L_c$/nm) | Aromaticity($f_a$) |
|---|---|---|---|---|
| anthracite coal | 0.3513 | 1.79 | 2.23 | 0.80 |

aromatic rings in the coal samples, that is, the size of the aromatic carbon mesh, where the sharper the peak resulted in a corresponding increase in the value of the representative parameter $L_a$. In addition, its condensation effect intensified, and the average diameter of aromatic carbon mesh also increased.

The original data were smoothed by Origin software, and then the 002 diffraction peaks of anthracite were fitted by peaks, where the 002 peaks with a regular arrangement were obtained for the analysis of microcrystalline structure parameters, and the fitting results are shown in Fig 8B. The microcrystalline structure parameters of anthracite could be obtained by taking the fitting results into Formula (2), as shown in Table 3.

## 3.4 Solid-state $^{13}$C nuclear magnetic resonance spectroscopy

The carbon structure in coal is very complex, and the peaks in the $^{13}$C NMR spectrum will be superimposed. Thus, it is necessary to simulate the peak separation of the spectrum to obtain the carbon functional groups and their relative content corresponding to a specific chemical shift. Origin software was used to fit the peak separation of the −50–200 ppm chemical shift, and the fitting results are shown in Fig 9.

According to the fitting results, there were five carbon signal peaks from left to right, namely, the methyl peak, oxygenated lipid carbon peak, protonated aromatic carbon peak, carboxyl peak, and carbonyl peak. According to the chemical shift, peak position, and content percentage of the different carbon atoms, 12 specific structural parameters of anthracite coal dust structure could be obtained, as shown in Table 4.

As shown by the $^{13}$C NMR structural parameters of anthracite in Table 3, the $f_{al}$:$f_a$:$f_a^C$ of anthracite was about 17:83:3, indicating that aromatic carbon was the main structural component in the anthracite molecules. The content of $f_a^H$ was higher than $f_a^N$, indicating that the aromatic ring was mainly provided by the protonated aromatic carbon, and the content of bridging the aromatic carbon was the highest in the non-protonated aromatic carbon. The $f_a^B$ content was 22.4%, indicating that the bridging aromatic carbon around the aromatic nucleus was the highest in the non-protonated aromatic carbon. The oxygen-containing functional groups in the macromolecular structure of coal were mainly a small amount of $f_{al}^O$ and $f_a^P$, and the low oxygen content was mainly caused by the loss of fat side chains and oxygen-containing functional groups in coal under coalification. The above structural parameters were consistent with the structural characteristics of anthracite.

Based on the fat carbon $f_{al} = f_{al}^* + f_{al}^H + f_{al}^O$ and aromatic carbon $f_a^{'} = f_a^H + f_a^N = f_a^H + f_a^P + f_a^S + f_a^B$, combined with Formula (3), the carbon ratio around the bridge of anthracite could be calculated to be 0.38, indicating that the aromatic structure of anthracite had a high degree of condensation.

## 4. The anthracite coal molecular model and model validation

### 4.1 The anthracite coal molecular model

The ratio of bridge carbon to peripheral carbon of anthracite was 0.38; thus, in the construction of a molecular model of anthracite, the aromatic carbon structure was dominated by naphthalene and anthracene, supplemented by pyrene and pentacene. Through MATLAB

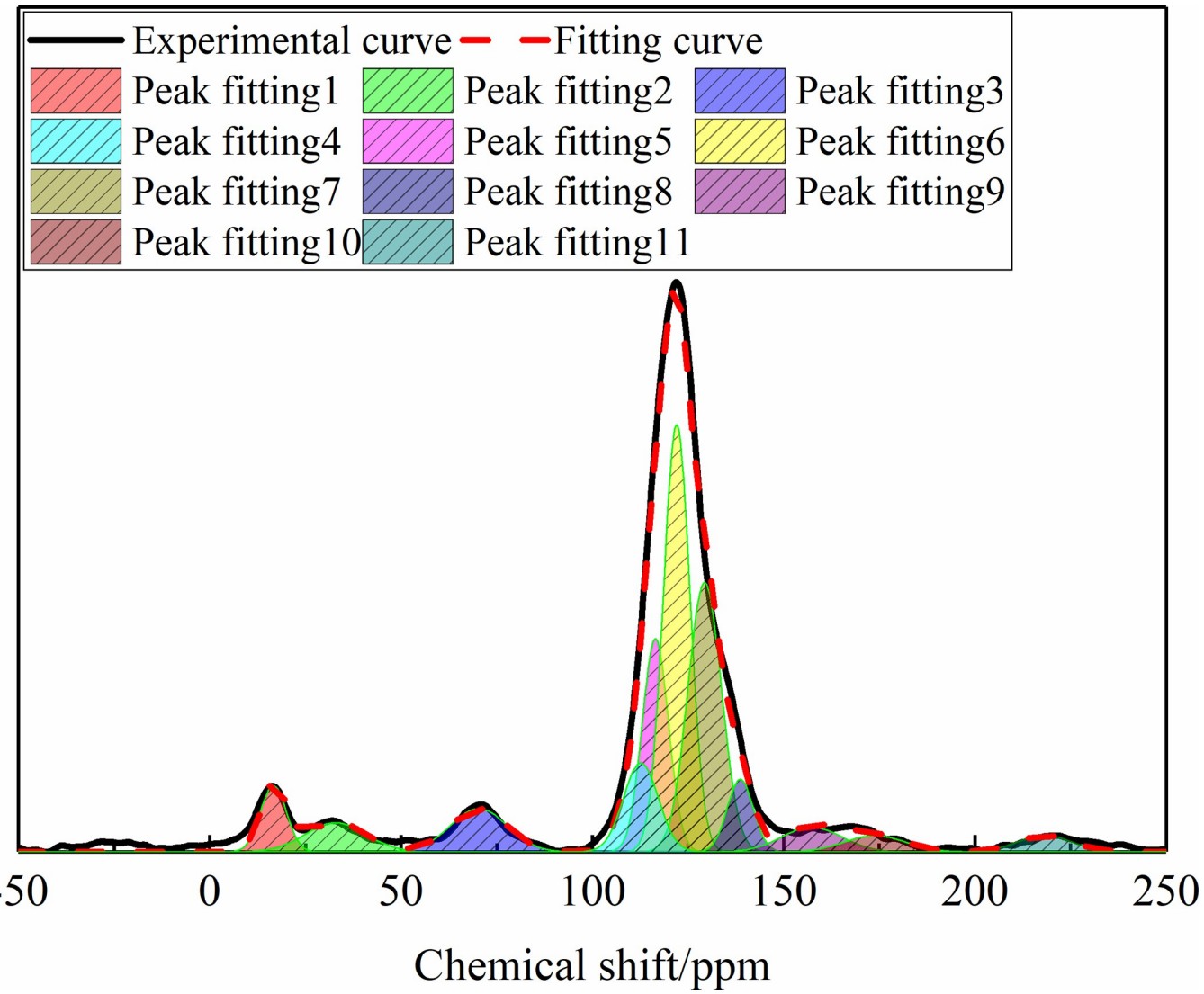

**Fig 9. 13C NMR peak fitting spectra of anthracite.**

programming calculations, the type and number of aromatic structural units closest to the bridge-to-cycle ratio in the experimental data in the molecular structure model of anthracite were obtained, and the aromatic skeleton combination in the structural model was finally determined, as shown in Table 5. At this time, the total number of aromatic ring carbons in the model was 168. According to $^{13}$C NMR, aromatic carbon accounted for 80.47%. Therefore, the total number of carbons in the molecular structure of anthracite was 208, while the total number of aliphatic carbons and (carboxyl) carbonyl carbons in the molecular structure of anthracite was calculated to be 40. According to the results of elemental analysis of the coal samples, the carbon content of anthracite was 85.53%, the oxygen content was 6.53%, the

**Table 4. Structural parameters of the coal sample molecular structure model.**

| Structural parameters | $f_{al}^{*}$ | $f_{al}^{H}$ | $f_{al}^{O}$ | $f_{a}^{H}$ | $f_{a}^{B}$ | $f_{a}^{S}$ | $f_{a}^{P}$ | $f_{a}^{N}$ | $f_{a}^{C}$ | $f_{al}$ | $f_{a}$ | $f_{a}^{'}$ |
|---|---|---|---|---|---|---|---|---|---|---|---|---|
| Content(%) | 4.76 | 4.90 | 7.11 | 49.67 | 22.40 | 4.68 | 3.68 | 30.76 | 2.56 | 16.97 | 83.03 | 80.47 |

**Table 5. Existence form of aromatic carbon in the anthracite configuration.**

| Mode of occurrence | Pyridine | Pyrrole | Naphthalene | Anthracene | Pyrene | Pentaphenyl |
|---|---|---|---|---|---|---|
| Quantity | 2 | 1 | 3 | 6 | 1 | 1 |

nitrogen content was 1.98%, and the sulfur content was 0.44%. The number of oxygens in the molecular structure of anthracite was 12, and the number of nitrogen atoms was 4. Due to the low sulfur content, and because the number was less than one, the molecular structure of anthracite constructed in this work did not contain sulfur. According to the analysis results of the XPS experiment, the nitrogen elements in anthracite mainly existed in the form of pyridine nitrogen and pyrrole nitrogen, and the number ratio was about 5: 8. Therefore, the existing mode of nitrogen in anthracite in this paper was designed as two pyridine nitrogen and two pyrrole nitrogen. According to the analysis of the content of oxygen-containing functional groups in FT-IR, the ratio of phenol, alcohol, ether (C-O), and carboxyl; carbonyl (C = O) in the oxygen-containing functional groups of anthracite was about 17.4:1, and the content of carboxyl was low. Therefore, the number of carboxyls in the molecular model of anthracite was set to 0, and the number of carbonyls was set to 1. Combined with the $^{13}C$ NMR test, the ratio of oxygen substitution and oxygen grafting fat content was about 1:1.9, and four hydroxyls ($-OH$) and seven oxygen grafting fats in anthracite could be determined.

Based on the above analysis results, the anthracite molecular model was built using Kingdraw chemical drawing software, and then the anthracite molecular model was imported into MestReNova software. By continuously adjusting the position and connection mode of the various groups in the anthracite molecular model, the aromatic units and aromaticity were kept unchanged. Finally, the predicted spectra of the model were compared with the experimental $^{13}C$ NMR spectra, as shown in Fig 10. The results showed that the $^{13}C$ NMR spectra of the established anthracite molecular model were in good agreement with the $^{13}C$ NMR experimental test spectra, which reflected the molecular structure of the coal sample. Finally, the molecular formula of Yangquan anthracite was determined as $C_{208}H_{162}O_{12}N_4$ (C:85.86%, N:1.93%, O:6.60%, H:5.61%, which was close to the test results of ultimate analysis in the coal samples). The molecular model of anthracite is shown in Fig 11A.

The 2D planar molecular model of anthracite constructed above was imported into Materials Studio 2019 (MS) molecular simulation software. After adding H saturation, the Forcite module was used to perform multiple geometric optimizations, annealing treatment, and kinetic treatment on the molecular structure of anthracite. The COMPASS force field was selected, and the calculation accuracy was set to fine. The charges term was set as the forcefield assignment, and the NPT kinetic ensemble (300–600K, five cycles) was used, where the maximum number of iterations was 2000. After multiple optimization treatments, the molecular low-energy structure of anthracite was finally obtained, as shown in Fig 11B. The $d_{002}$ = 0.3463 nm, $L_c$ = 2.0037 nm, and $L_a$ = 1.5754 nm values of anthracite were measured using the measure tool in MS software, which was close to the $d_{002}$ = 0.3513 nm, $L_c$ = 2.23 nm, and $L_a$ = 1.79 nm values measured by XRD experimentation, further indicating that the molecular 3D model of anthracite was reasonable.

## 4.2 The anthracite coal molecular model validation

To further verify the rationality of the parameters of the molecular model of anthracite, the molecular structure cell model of anthracite was established, and the adsorption of $CH_4$ gas molecules in the molecular structure model of anthracite was observed [34–36]. The molecular structure models of 15 anthracites were obtained, and the Amorphous Cell module was used.

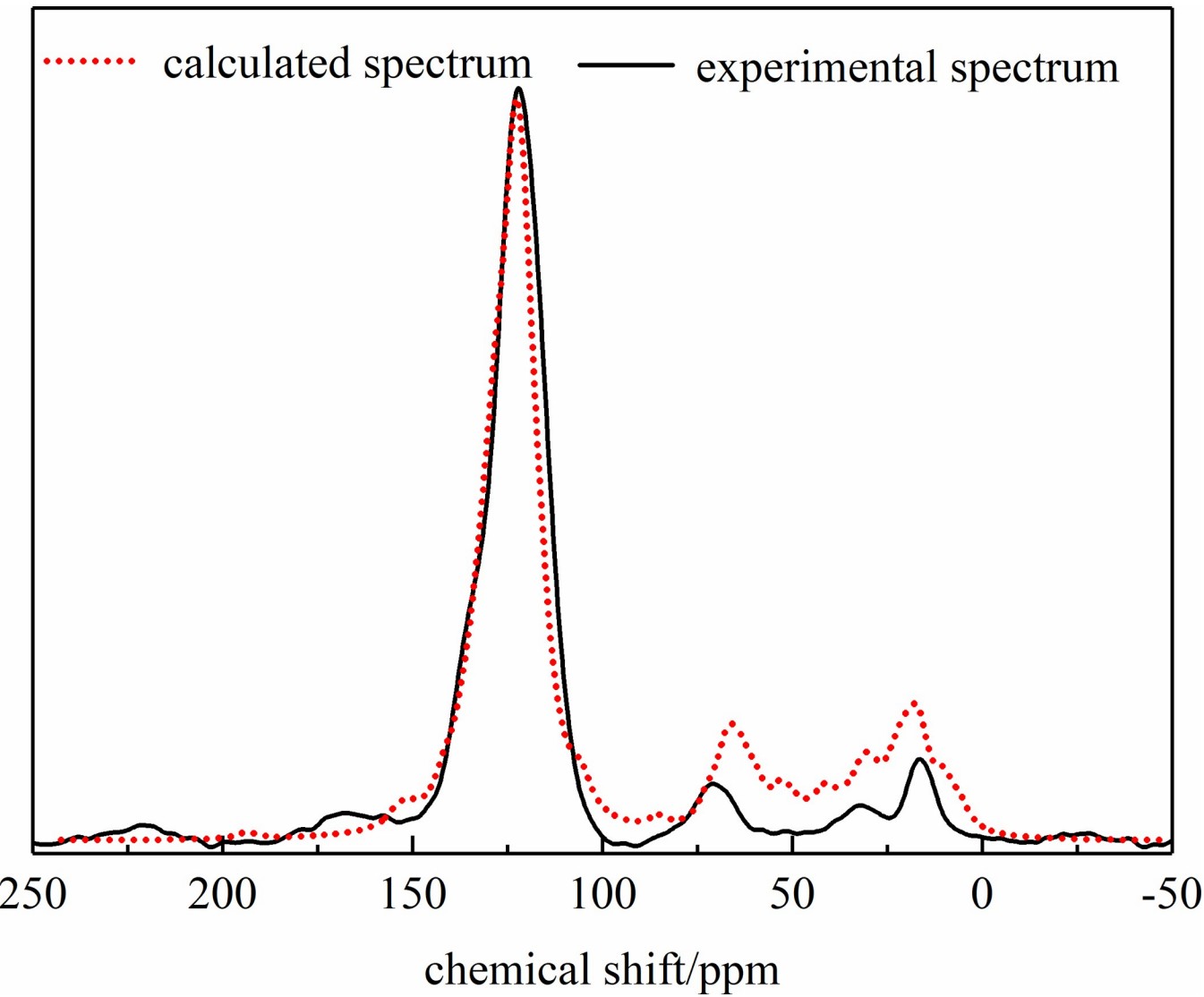

**Fig 10. Comparison of 13C NMR experimental spectra and model predictive spectra of anthracite.**

The calculation accuracy was fine, and we used the COMPASS force field. The 15 single molecular structures were placed into the cell to add three-dimensional periodic boundary conditions, and the density was set to 1.32 g/cm$^3$. Then, structure optimization and dynamic optimization of the cell model of anthracite were carried out to minimize and stabilize the energy of the constructed coal molecular structure model. Finally, the structure model size of anthracite with low energy conformation was $A = B = C = 3.89034$ nm, where the molecular formula was $C_{3120}H_{2430}N_{60}O_{180}$. The cell model of the anthracite macromolecular structure is shown in Fig 12.

MS software was used to analyze the adsorption of CH4 gas molecules in the molecular structure model of anthracite by combining GCMC and MD. The simulation process was completed by the adsorption and Forcite modules in MS. The adsorption data of the $CH_4$ molecules in the anthracite molecular model were compared with the experimental data of anthracite in reference 17, as shown in Fig 13. The comparison showed that the adsorption amount of $CH_4$ gas was within an order of magnitude, which proved that the anthracite molecular

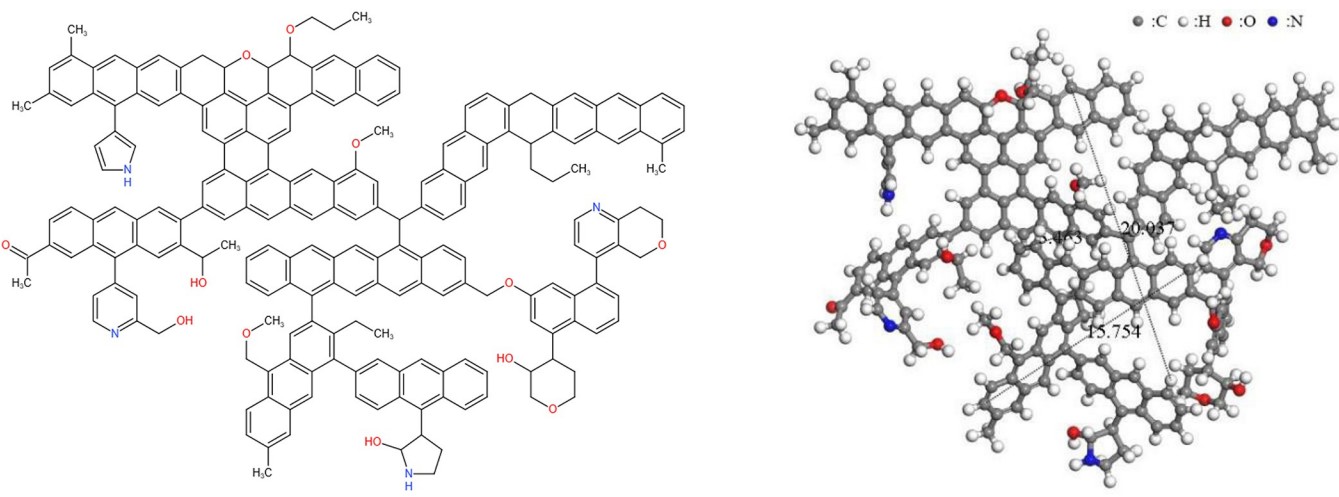

**Fig 11. Molecular model of anthracite coal.** (a) 2D structure, (b) 3D optimized structure.

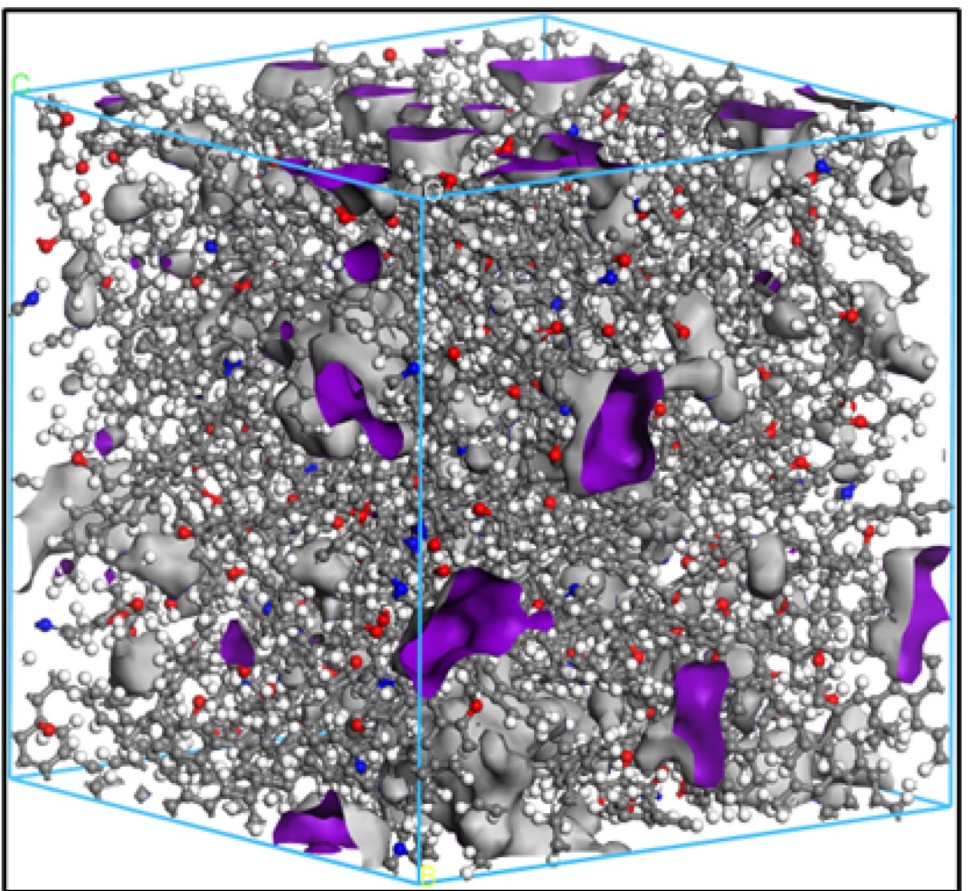

**Fig 12. Cellular model of the molecular structure of anthracite.**

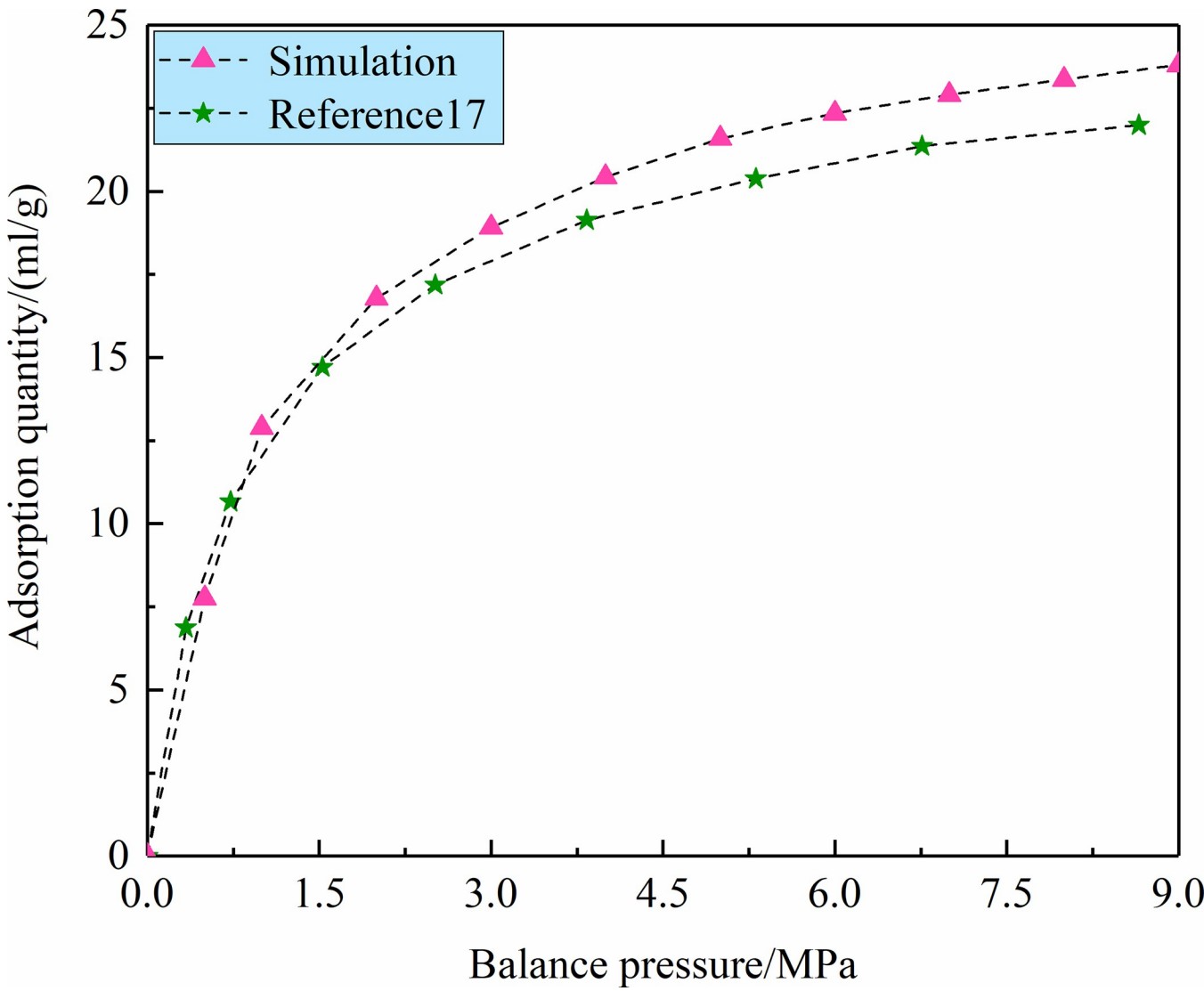

**Fig 13. Comparison of CH4 adsorption simulation and experiment.**

model established in this work conformed to the structural characteristics of anthracite. However, the columnar coal samples used in the adsorption experiment had defects to some extent, resulting in a decrease in the pore volume and specific surface area in the coal body, which reduced the gas adsorption capacity in the experimental process. The variation trend of the $CH_4$ adsorption amount obtained by the experiments and molecular simulation was basically consistent, which further proved that the anthracite molecular model established in this work was reasonable.

## 5. Conclusions

Through a series of analyses of Yangquan anthracite, information such as the structure and functional groups could be obtained in this study. The main conclusions were as follows.

(1) The content information of the different elements in anthracite was obtained by industrial and element analyses.

(2) FT-IR was used to reveal the composition of the functional groups, and the number of different substituted benzene rings in the anthracite molecular structure was quantitatively analyzed, which provided a theoretical basis for the accurate construction of the coal macromolecular structure.

(3) XPS analysis showed that nitrogen was mainly in the form of pyridine nitrogen and pyrrole nitrogen, and sulfur was mainly in the form of thiophene.

(4) The microcrystalline structure parameters in anthracite were obtained by XRD analysis, and the conclusion verified that the three-dimensional model structure was reasonable.

(5) $^{13}$C NMR analysis showed that the aromatic ring in anthracite had a high degree of condensation, and the ratio of bridge carbon to peripheral carbon was 0.38.

According to the above analysis results, the molecular plane structure model and 3D structure model of Yangquan anthracite coal dust were constructed, and the molecular formulas were $C_{208}H_{162}O_{12}N_4$ and $C_{3120}H_{2430}N_{60}O_{180}$. the predicted spectrum verified that the models represented the real coal molecule. The adsorption data of the $CH_4$ molecule in the anthracite molecular model were basically consistent with the experimental data of anthracite, which further proved that the anthracite molecular model built in this work was reasonable. The adsorption data of the $CH_4$ molecules on the anthracite molecular model were basically consistent with the experimental data of anthracite in reference 17, which further proved that the anthracite molecular model established in this paper was reasonable. This work provided a method to deeply understand the structural characteristics of anthracite and establish its structural model.

## Author Contributions

**Conceptualization:** Jinzhang Jia, Yumo Wu.

**Data curation:** Jinzhang Jia, Yumo Wu, Bin Li.

**Formal analysis:** Yumo Wu, Bin Li.

**Funding acquisition:** Jinzhang Jia.

**Investigation:** Jinzhang Jia, Yumo Wu, Dan Zhao.

**Methodology:** Jinzhang Jia, Yumo Wu.

**Software:** Yumo Wu, Dan Zhao, Bin Li.

**Validation:** Jinzhang Jia, Dan Zhao, Dongming Wang.

**Visualization:** Yumo Wu, Bin Li.

**Writing – original draft:** Jinzhang Jia, Yumo Wu, Dan Zhao, Bin Li.

**Writing – review & editing:** Yumo Wu, Dan Zhao, Fengxiao Wang, Yinuo Chen.

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
