## [Decision Letter · Decision Letter 0]

1 Sep 2022

PONE-D-22-16377Molecular structure
characterization analysis and molecular model construction of
anthracitePLOS ONE

Dear Dr. WU,

Thank you for submitting your manuscript to PLOS ONE. After careful consideration, we
feel that it has merit but does not fully meet PLOS ONE’s publication criteria as it
currently stands. Therefore, we invite you to submit a revised version of the
manuscript that addresses the points raised during the review process.

Please submit your revised manuscript by Oct 16 2022 11:59PM. If you will need more
time than this to complete your revisions, please reply to this message or contact
the journal office at plosone@plos.org. When
you're ready to submit your revision, log on to https://www.editorialmanager.com/pone/ and select the 'Submissions
Needing Revision' folder to locate your manuscript file.

Please include the following items when submitting your revised
manuscript:A rebuttal letter that responds to each point raised by the academic
editor and reviewer(s). You should upload this letter as a separate file
labeled 'Response to Reviewers'.A marked-up copy of your manuscript that highlights changes made to the
original version. You should upload this as a separate file labeled
'Revised Manuscript with Track Changes'.An unmarked version of your revised paper without tracked changes. You
should upload this as a separate file labeled 'Manuscript'.If you would like to make changes to your financial disclosure,
please include your updated statement in your cover letter. Guidelines for
resubmitting your figure files are available below the reviewer comments at the end
of this letter.

We look forward to receiving your revised manuscript.

Kind regards,

Saraboji Kadhirvel, Ph.D

Academic Editor

PLOS ONE

Journal Requirements:

“YES - Specify the role(s) played.

This work was partly supported by the National Natural Science Foundation of China
(grant number 52174183)（http://www.nsfc.gov.cn/）, and the Natural Science Foundation of
Liaoning Province (grant number 2019-MS-162)（http://kjt.ln.gov.cn/）.”

Reviewers' comments:

Reviewer's Responses to Questions

**Comments to the Author**

1. Is the manuscript technically sound, and do the data support the conclusions?

Reviewer #1: Yes

Reviewer #2: Yes

2. Has the statistical analysis been performed
appropriately and rigorously? 

Reviewer #1: Yes

Reviewer #2: Yes

3. Have the authors made all data underlying the
findings in their manuscript fully available?

Reviewer #1: Yes

Reviewer #2: Yes

4. Is the manuscript presented in an intelligible
fashion and written in standard English?

Reviewer #1: Yes

Reviewer #2: Yes

5. Review Comments to the Author

Reviewer #1: Interesting work. Recommend for publication in the journal, PLOS ONE
after mandatory minor revision.

(i) x-axis of IR spectra is wrong

(ii) y-axis of XRD pattern is wrong

(iii) improve the quality of figures

Reviewer #2: Accept with minor revision.

First, I appreciate the authors for their initiative to understand the structure of
anthracite. Second, I also appreciate that the authors have given due to credit to
the previous work. Third, the authors have demonstrated their skills on FT-IR and
using which they have analysed the structure in a very well manner. Finally, the
conclusion has reflected their analysis and the manuscript is in a proper format to
understand the content very well.

Questions to Authors:

1. The authors have used elemental analysis to understand the CHNO content C208 H162
O12 N4 is matching with the proposed structure (I took the pain of drawing the whole
structure to compare the molecular structure). however, the ultimate analysis shows
very less H content (85.43:4.52:6.53:1.98). why these discrepancies? and also please
shine a light on proximate and ultimate analysis.

2. Kindly add the following reference “New advances in coal Structural model”
Zhihong, Q., International journal of mining science and technology, 2018,
541-559.

6. PLOS authors have the option to publish the peer
review history of their article (what does this mean?). If published, this will
include your full peer review and any attached files.

If you choose “no”, your identity will remain anonymous but your review may still be
made public.

**Do you want your identity to be public for this peer review?** For
information about this choice, including consent withdrawal, please see our
Privacy Policy.

Reviewer #1: **Yes: **Devaraj S

Reviewer #2: No

---

## [Author Response · Author response to Decision Letter 0]

8 Sep 2022

Dear editor: 

Thank you very much for giving us an opportunity to revise our manuscript. We
appreciate the editor and reviewers very much for their constructive comments and
suggestions on our manuscript entitled“Molecular structure characterization analysis
and molecular model construction of anthracite”(ID: PONE-D-22-16377).

We have studied reviewers' comments carefully. According to the reviewers' detailed
suggestions. we have made a careful revision on the original manuscript. I modified
the copy of the markup to form a separate file, named 'Revised Manuscript with Track
Changes'.

Kind regards.

Corresponding author: WU Yumo

E-mail address: 13614067811@163.com

Replies to the reviewers’ comments: 

Reviewer #1:

1. (i) x-axis of IR spectra is wrong

(ii) y-axis of XRD pattern is wrong

(iii) improve the quality of figures

Response: We feel sorry for our carelessness. In our resubmitted manuscript, the typo
is revised, and upload the modified image as a separate file. Thanks for your
correction.

Reviewer #2:

1. The authors have used elemental analysis to understand the CHNO content C208 H162
O12 N4 is matching with the proposed structure (I took the pain of drawing the whole
structure to compare the molecular structure). however, the ultimate analysis shows
very less H content (85.43:4.52:6.53:1.98). why these discrepancies? and also please
shine a light on proximate and ultimate analysis.

Response: We sincerely thank the reviewer for careful reading. In the structural
model, the red-labeled H is only shown in the hydroxyl group, while the remaining H
is not shown in the benzene ring and other structures.

2. Kindly add the following reference “New advances in coal Structural model”
Zhihong, Q., International journal of mining science and technology, 2018, 541-559. 

Response: We sincerely appreciate the valuable comments. As suggested by the
reviewer, we have added more references to support this idea (New advances in coal
Structural model). According to literature 36, our model building has played a
guiding role.

to Reviewers.docx
---

## [Editor Report · Decision Letter 1]

11 Sep 2022

Molecular structure characterization analysis and molecular model construction of
anthracite

PONE-D-22-16377R1

Dear Dr. WU,

We’re pleased to inform you that your manuscript has been judged scientifically
suitable for publication and will be formally accepted for publication once it meets
all outstanding technical requirements.

Kind regards,

Saraboji Kadhirvel, Ph.D

Academic Editor

PLOS ONE
---

## [Editor Report · Acceptance letter]

13 Sep 2022

PONE-D-22-16377R1 

Molecular structure characterization analysis and molecular model construction of
anthracite 

Dear Dr. Wu:

I'm pleased to inform you that your manuscript has been deemed suitable for
publication in PLOS ONE. Congratulations! Your manuscript is now with our production
department. 

Kind regards, 

on behalf of

Dr. Saraboji Kadhirvel 

Academic Editor

PLOS ONE